# Discovery of a Novel Simian Pegivirus in Common Marmosets (*Callithrix jacchus*) with Lymphocytic Enterocolitis

**DOI:** 10.3390/microorganisms8101509

**Published:** 2020-09-30

**Authors:** Anna S. Heffron, Michael Lauck, Elizabeth D. Somsen, Elizabeth C. Townsend, Adam L. Bailey, Megan Sosa, Jens Eickhoff, Saverio Capuano III, Christina M. Newman, Jens H. Kuhn, Andres Mejia, Heather A. Simmons, David H. O’Connor

**Affiliations:** 1Department of Pathology and Laboratory Medicine, School of Medicine and Public Health, University of Wisconsin-Madison, Madison, WI 53711, USA; heffron2@wisc.edu (A.S.H.); michaellauck@gmail.com (M.L.); esomsen@wisc.edu (E.D.S.); liz.townsend@wisc.edu (E.C.T.); ccondon2@wisc.edu (C.M.N.); 2Department of Pathology and Immunology, Washington University School of Medicine, St. Louis, MO 63110, USA; adam.lee.bailey@gmail.com; 3Wisconsin National Primate Research Center, University of Wisconsin-Madison, Madison, WI 53715, USA; msosa@primate.wisc.edu (M.S.); capuano@primate.wisc.edu (S.C.III); amejia@primate.wisc.edu (A.M.); hsimmons@primate.wisc.edu (H.A.S.); 4Department of Biostatistics & Medical Informatics, University of Wisconsin-Madison, Madison, WI 53705, USA; eickhoff@biostat.wisc.edu; 5Integrated Research Facility at Fort Detrick, National Institute of Allergy and Infectious Diseases, National Institutes of Health, Fort Detrick, Frederick, MD 21702, USA; kuhnjens@niaid.nih.gov

**Keywords:** pegivirus, flavivirus, *Callithrix jacchus*, common marmoset, lymphocytic enterocolitis, next-generation sequencing, novel virus discovery

## Abstract

From 2010 to 2015, 73 common marmosets (*Callithrix jacchus*) housed at the Wisconsin National Primate Research Center (WNPRC) were diagnosed postmortem with lymphocytic enterocolitis. We used unbiased deep-sequencing to screen the blood of deceased enterocolitis-positive marmosets for viruses. In five out of eight common marmosets with lymphocytic enterocolitis, we discovered a novel pegivirus not present in ten matched, clinically normal controls. The novel virus, which we named Southwest bike trail virus (SOBV), is most closely related (68% nucleotide identity) to a strain of simian pegivirus A isolated from a three-striped night monkey (*Aotus trivirgatus*). We screened 146 living WNPRC common marmosets for SOBV, finding an overall prevalence of 34% (50/146). Over four years, 85 of these 146 animals died or were euthanized. Histological examination revealed 27 SOBV-positive marmosets from this cohort had lymphocytic enterocolitis, compared to 42 SOBV-negative marmosets, indicating no association between SOBV and disease in this cohort (*p* = 0.0798). We also detected SOBV in two of 33 (6%) clinically normal marmosets screened during transfer from the New England Primate Research Center, suggesting SOBV could be exerting confounding influences on comparisons of common marmoset studies from multiple colonies.

## 1. Introduction

Common marmosets (*Callithrix jacchus*) are a valuable model species due to their small body size, communal monogamous familial behavior, birth of hematopoietic chimeric litters, short parturition intervals, and status as members of a non-endangered primate species [1,2,3,4,5]. The utility of common marmosets in research resulted in a recent increase in demand for these animals [6]. The Wisconsin National Primate Research Center (WNPRC) in Madison, Wisconsin, USA, houses a common marmoset colony typically consisting of about 240 common marmosets, which are used by researchers at the University of Wisconsin-Madison for groundbreaking research in neurological, neurobehavioral, and pharmacologic research, among many others [7,8,9,10,11,12,13,14,15,16,17].

From 2010 to 2015, 73 common marmosets housed at the WNPRC were euthanized due experimental end point, chronic intractable diarrhea, or chronic severe weight loss; they then underwent necropsy with histology and were diagnosed with lymphocytic enterocolitis [18,19,20,21]. Beyond the regrettable loss of animal life, common marmoset morbidity and mortality due to enterocolitis is harmful both to colony success and to the scientific studies to which these animals are assigned. Though lymphocytic enterocolitis is one of the most common causes of death in captive common marmosets [18,19,20,21,22,23], the epizootic at the WNPRC was associated with an unusually high disease incidence for the colony, prompting investigations into a possible infectious contributor. Unbiased deep-sequencing led to the discovery of two similar variants of a novel pegivirus, most closely related to a variant of simian pegivirus A (SPgV-A) previously isolated from a three-striped night monkey (*Aotus trivirgatus*). This novel pegivirus was present in a subset of deceased common marmosets diagnosed postmortem with lymphocytic enterocolitis and was not present in matched, clinically normal controls. 

Pegiviruses, members of genus *Pegivirus (Amarillovirales*: *Flaviviridae)*, are ubiquitous in animal populations [24,25,26,27,28,29,30,31,32,33,34,35], but their biological consequences are poorly understood. Pegiviruses can persist at high titers for years or decades in humans [36,37,38,39,40] and chimpanzees [41] with an unusually low mutation rate compared to other RNA viruses [38,42], and they have never been shown to be the causative agent of any disease [43,44,45,46,47,48,49,50,51,52,53,54,55,56,57,58,59,60]. Apparent links between pegiviruses and disease, such as that initially posited for Theiler’s disease-associated virus (TDAV) and Theiler’s disease [61,62], have later been shown to be more likely spurious [35,63]. The mechanisms of pegivirus biology have eluded definition, but these viruses are considered most likely lymphotropic [64,65,66,67], and evidence from in vivo and in vitro studies suggests they may affect T cell functioning and homeostasis [68,69,70,71,72,73,74,75,76]. Lymphocytic enterocolitis in common marmosets is likewise characterized by a dysregulation of T cell biology, as the intestinal villus architecture is disrupted or lost due to the intraepithelial infiltration of large numbers of CD3 CD8-positive lymphocytes [22]. Given the importance of common marmosets as a model species and the disease burden caused by lymphocytic enterocolitis, we set out to characterize the possible link between this new virus and the disease state.

Here, we report the discovery of two variants of a novel pegivirus in a captive common marmoset colony. We establish phylogenetic relationships with other known pegiviruses. Since this virus was discovered in common marmosets with lymphocytic enterocolitis and was absent in clinically normal controls, we measure the prevalence of the virus in the colony and track the potential association of viral status with risk of developing lymphocytic enterocolitis disease over four years. Our findings have implications for animal studies in which specific pathogen-free animals are desired, and they demonstrate the need for further investigations to increase understanding of these viruses and their impact on common marmoset health.

## 2. Materials and Methods 

### 2.1. Animals

All animals in this study were common marmosets (*Callithrix jacchus* Linnaeus, 1758) housed at the Wisconsin National Primate Research Center (WNPRC) in Madison, WI, USA. The common marmoset colony at the WNPRC was established in 1960. The original animals were imported from northeastern Brazil, with the final importation occurring in the early 1970s. The average yearly population of the colony each year from 2010 to 2019 was approximately 240 animals, all of which were born in captivity. WNPRC animals screened were 41% (60 animals) female and 59% (86 animals) male. Age at the time of screening ranged from 0.82–12.82 years (mean 4.65 +/- 2.83 years, median 4.26 years).

The New England Primate Research Center (NEPRC), Southborough, MA, USA, was closed in 2015, resulting in a transfer of 82 common marmosets to WNPRC before closure in November 2014. Plasma samples were collected from 33 of these animals upon their arrival at WNPRC (November–December 2014) while quarantined in a separate building and location from the WNPRC marmoset colony. In the population initially from the NEPRC, 16 (48%) of the screened animals were female, and 17 (52%) were male. Age at the time of screening ranged from 0.66–9.42 years (mean 4.21 ± 2.87, median 3.76 years) in this population.

### 2.2. Ethics

All common marmosets were cared for by WNPRC staff according to the regulations and guidelines outlined in the National Research Council’s Guide for the Care and Use of Laboratory Animals, the Animal Welfare Act, the Public Health Service Policy on the Humane Care and Use of Laboratory Animals, and the recommendations of the Weatherall report (https://royalsociety.org/topics-policy/publications/2006/weatherall-report/). Per WNPRC standard operating procedures for animals assigned to protocols involving the experimental inoculation of infectious pathogens, environmental enhancement included constant visual, auditory, and olfactory contact with conspecifics, the provision of feeding devices that inspire foraging behavior, the provision and rotation of novel manipulanda, and enclosure furniture (i.e., perches, shelves). The common marmosets were housed socially in enclosures measuring 0.6 mD × 0.9 mW × 1.8 mH or 0.6 mD × 1.2 mW × 1.8 mH. The WNPRC maintains an exemption from the USDA for these enclosures as they do not meet the Animal Welfare Act regulations for floor space but greatly exceed height requirements as the species are arboreal. This study was approved by the University of Wisconsin-Madison College of Letters and Sciences and Vice Chancellor for Research and Graduate Education Centers Institutional Animal Care and Use Committee (animal protocol numbers G005401 and G005443).

### 2.3. Unbiased Deep-Sequencing

Samples from 18 common marmosets (eight deceased common marmosets diagnosed with lymphocytic enterocolitis through necropsy and 10 live, healthy common marmosets) from the WNPRC and 12 common marmosets (all live and healthy) from the NEPRC were screened for the presence of viruses using unbiased deep-sequencing. The live WNPRC common marmosets and the live NEPRC common marmosets were selected randomly for deep-sequencing.

DNA and RNA were isolated from plasma. Common marmoset plasma (1 mL/animal) was centrifuged at 5000× *g* for 5 min at 4 °C. Supernatants were removed and filtered through a 0.45 µm filter, then centrifuged at maximum speed (20,817× *g*) for 5 min at 4 °C. Supernatants were removed and incubated for 90 min at 37 °C with a DNA/RNA digest cocktail consisting of 4 μL DNAfree DNAse (0.04 U/μL; Ambion, Austin, TX, USA), 6 μL Baseline Zero DNAse (0.1 U/μL, Epicentre Technologies, Madison, WI, USA), 1 μL Benzonase (1 U/μL, Sigma-Adrich, St. Louis, MO, USA), and 12 μL DNAse 10x buffer. Viral nucleic acids were then isolated using the Qiagen QIAamp MinElute Virus Spin Kit without the use of AW1 buffer or carrier RNA (Qiagen, Valencia, CA, USA). Random hexamers were used to prime cDNA synthesis (Life Technologies, Grand Island, NY, USA), followed by DNA purification using Ampure XP beads, as previously described [77,78]. Deep-sequencing libraries were prepared using the Nextera XT DNA Library Prep Kit (Illumina, San Diego, CA, USA) and sequenced on MiSeq (Illumina). 

### 2.4. Viral Sequence and Phylogenetic Analysis

Sequence data were analyzed using CLC Genomics Workbench 5.5 (CLC bio, Aarhus, Denmark). Low-quality reads (Phred < Q30) and short reads (<100 bp) were removed with CLC Genomics Workbench 7.1 (CLC bio, Aarhus, Denmark), and the remaining reads were assembled de novo using the MEGAHIT assembler. Assembled contiguous sequences (contigs) and singleton reads were queried against the GenBank nucleotide database using the basic local alignment search tools blastn. Nucleotide sequences were codon aligned individually for all known pegiviruses with complete genomes using ClustalW2 in the alignment editor program in MEGA6.06 and edited manually. The best-fitting distance model of nucleotide substitution for each alignment was inferred using the maximum likelihood (ML) method with goodness of fit measured by the Bayesian information criterion in MEGA6.06. The best-fitting nucleotide substitution model for the phylogenetic alignments was inferred to be the GTR model with discrete gamma and invariant among-site rate variation. The GenBank accession numbers of the sequences used are MT513216 (Southwest bike trail virus variant 1 [SOBV-1]), MT513217 (Southwest bike trail virus variant 2 [SOBV-2]), MK059751 (dolphin pegivirus), KU351669 (porcine pegivirus), KC815311 (rodent pegivirus), KC796088 (bat pegivirus I), KT439329 (human hepegivirus), KC796076 (bat pegivirus G), KC796080 (bat pegivirus F), KC410872 (equine pegivirus), KC145265 (Theiler’s disease-associated virus), U44402 (human pegivirus genotype 2), GU566734 (GB virus-D), U22303 (simian pegivirus A isolated from black-mantled tamarins [*Saguinus nigricollis*]), AF023425 (simian pegivirus A isolated from three-striped night monkeys [*Aotus trivirgatus*]), AF023424 (simian pegivirus A isolated from mustached tamarins [*S. mystax*]), NC_001837 (simian pegivirus A isolated from white-lipped tamarins [*S. labiatus*]), KC796075 (bat pegivirus PDB737B), KC796075 (bat pegivirus PDB106), KC796082 (bat pegivirus PDB24), U63715 (human pegivirus genotype 1), D87713 (human pegivirus genotype 3), AB021287 (human pegivirus genotype 4), AY949771 (human pegivirus genotype 5), AB003292 (human pegivirus genotype 6), and AF070476 (simian pegivirus A isolated from common chimpanzees [*Pan troglodytes*]).

Protein family analysis and putative protein predictions were performed using Pfam (http://pfam.xfam.org/). The amino acid similarity of the novel pegivirus with related pegivirus lineages was determined across the polyprotein using SimPlot v3.5.1 [79] following TranslatorX alignment (MAAFT) without Gblocks cleaning. The GenBank accession numbers of the sequences used are HGU22303 (GBV-A-like virus recovered from black-mantled tamarins), AF023425 (GBV-A-like virus recovered from three-striped night monkeys), AF023424 (GBV-A-like virus recovered from mustached tamarins), NC_001837 (GBV-A-like virus recovered from white-lipped tamarins), and KC796081 (bat pegivirus recovered from African straw-colored fruit bats [*Eidolon helvum*]).

The sequence similarity matrix was created in Geneious Prime 2020.1.2 (Auckland, New Zealand) using representative members of each pegivirus species [80,81].

### 2.5. Screening for SOBV by RT-PCR

Plasma samples from 136 healthy WNPRC common marmosets were screened specifically for SOBV by RT-PCR. Twenty plasma samples collected from NEPRC animals were likewise screened by RT-PCR.

Screening of these animals was performed with samples from animals positive for SOBV by deep sequencing as positive controls. RNA was isolated from 100–500 μL of plasma using the QIAamp Viral RNA Mini Kit (Qiagen). A primer set (forward primer: GGTGGTCCACGAGTGATGA; reverse primer: AGGTACCGCCTGGGGTTAG) targeting a region of the viral helicase which was conserved among the animals initially positive by deep-sequencing was designed, resulting in a 615-bp amplicon. Viral RNA was reverse-transcribed and amplified using the SuperScript III High Fidelity One-Step RT-PCR kit (Invitrogen, Life Technologies, Carlsbad, CA, USA). The reverse transcription-PCR conditions were as follows: 50 °C for 30 min; 94 °C for 2 min; 40 cycles of 94 °C for 15 s, 55 °C for 30 s, and 68 °C for 1 min; and 68 °C for 5 min. Following PCR, amplicons were purified from excised gel slices (1% agarose) using the Qiagen MinElute Gel Extraction kit (Qiagen). Each amplicon was quantified using Quant-IT HS reagents (Invitrogen), and approximately 1 ng of each was used in a tagmentation reaction with the Nextera XT DNA Library Prep Kit. Final libraries representing each amplicon were characterized for average length using a DNA high sensitivity chip on a 2100 bioanalyzer (Agilent Technologies, Loveland, CO, USA) and quantitated with Quant-IT HS reagents. Libraries were sequenced on a MiSeq.

### 2.6. Post Mortem Diagnosis of Lymphocytic Enterocolitis

All animals humanely euthanized or that die spontaneously at the WNPRC undergo complete postmortem examinations (necropsy) with the collection of a standardized set of tissues for histologic evaluation and ancillary diagnostics (if necessary). Lymphocytic enterocolitis is well-characterized in common marmosets [22] and a marmoset-specific enteric collection has been established for all marmoset necropsies. Hematoxylin and eosin (H&E) stains are used for histological examination to evaluate for spontaneous diseases and experimentally induced tissue and organ changes. In this study, immunohistochemical (IHC) CD3 and CD20 or CD79 staining was additionally performed on samples from these animals to differentiate lymphocyte populations (primarily T cells, B cells, or mixed T and B cells). Diagnosis of T cell rich lymphocytic enterocolitis was based on abnormal architecture of the intestines and IHC staining [22,82]. If confounding factors hampered diagnosis (e.g., severe B cell lymphoma or autolysis), the data for that animal were removed from the analysis.

### 2.7. Statistical Analysis

We used univariate logistic regression to evaluate the associations of SOBV viremia with enterocolitis risk. Analyses were repeated to determine association with lymphocytic disease in small intestines only, large intestines only, both the small and large intestines, and either the small or large intestines. All reported *p*-values are two-sided and *p* < 0.05 was used to define statistical significance. Statistical analyses were conducted using R version 3.6.3 in RStudio version 1.1.383. 

### 2.8. Data Accessibility and Management

Metagenomic sequencing data have been deposited in the Sequence Read Archive (SRA) under Bioproject PRJNA613737. Derived data, analysis pipelines, and figures are available for easy replication of these results at a publicly accessible GitHub repository (https://github.com/aheffron/SPgVwnprc_in_marmosets).

## 3. Results

### 3.1. Captive Common Marmosets Harbor a Novel Pegivirus

To examine the etiology of the unusually high rate of lymphocytic enterocolitis in deceased WNPRC common marmosets, banked plasma samples from eight common marmosets diagnosed with lymphocytic enterocolitis and from ten clinically normal, live common marmosets to be used as controls were screened by deep-sequencing for the presence of viral RNA. RNA from a previously undocumented pegivirus was detected in the plasma of five of eight deceased marmosets with lymphocytic enterocolitis. We propose this novel virus (BioProject PRJNA613737) be formally named the Southwest bike trail virus (SOBV). Pegivirus RNA was not detected in the plasma of the ten clinically normal common marmoset controls.

SOBV consists of a 9.8-kb-long contig that is highly similar to the genome of simian pegivirus A (SPgV-A) trivirgatus, a simian pegivirus previously discovered in a three-striped night monkey (*Aotus trivirgatus*) [27] (Figure 1), with 68% nucleotide identity across the coding sequence when aligned using ClustalW with an IUB cost matrix (gap extension cost, 6.66; gap open cost, 15.00). Four of the five common marmosets positive for SOBV had variants of the virus with 98–99% sequence identity, while one common marmoset had a variant with 88% sequence identity to the others. We have named these variants SOBV-1 (GenBank accession number MT513216) and SOBV-2 (GenBank accession number MT513217).

Pairwise comparisons of amino acid identity across the entire coding region further illustrate the similarity of SOBV-1 and SOBV-2 and the divergence between these novel virus strains and the next most closely related viruses (Figure 2 and Figure 3), most of which were simian pegiviruses. Several pegivirus isolates found in a bat [83] also shared high degrees of similarity with the novel pegivirus.

### 3.2. Novel Pegivirus RNA Is Detected in up to 34% of a Captive Common Marmoset Colony

Having identified the novel pegivirus in diseased animals, we sought to determine its prevalence within the WNPRC common marmoset colony. We developed an RT-PCR assay to detect a conserved region of the putative helicase protein of SOBV and used this to screen plasma collected from 146 clinically normal live common marmosets in the WNPRC colony, confirming results through deep-sequencing of the amplicons. At the time of the initial screening in March–April 2014, 50 of the 146 (34.25%) clinically normal screened animals tested positive for SOBV RNA. Nineteen of 60 females (31.67%) and 31 of 86 males (36.05%) tested positive at the time of screening. Sex was not associated with the likelihood of SOBV using univariate logistic regression (*p* = 0.583). Age at the time of screening was associated with the likelihood of SOBV (*p* = 0.0324, odds ratio 1.144, 95% confidence interval 1.014–1.298), with the likelihood of positivity increasing with each additional year of age (Figure 4, Table 1).

In November 2014, 82 common marmosets were transferred from the New England Primate Research Center (NEPRC) to the WNPRC. Samples from 33 NEPRC common marmosets were collected while the animals were in quarantine. Two (6%) of these were found to be positive for SOBV RNA when screened by RT-PCR. 

### 3.3. Presence of Novel Pegivirus Is Not Statistically Significantly Associated with Lymphocytic Enterocolitis in the Common Marmoset 

Given that pegiviruses are known to persist in hosts for years or decades [36,37,38,39,40,41], we sought to determine whether SOBV-positive animals were more likely to develop lymphocytic enterocolitis over a period of observation. Typical enteric architecture consists of slender, often branching villi, with short intestinal glands, small numbers of lymphocytes in the lamina propria, and prominent B cell aggregates dispersed throughout the length of the intestines (Figure 5, control). Lymphocytic enterocolitis was diagnosed as a disruption of this architecture, with lymphocytic infiltration that expands the lamina propria, resulting in widening and shortening of villi and hyperplasia of crypt epithelium (Figure 5, E1–E3). Cases varied in severity, with mild cases showing only slight expansion of the lamina propria and advanced cases showing complete loss of villus architecture due to infiltration of the lamina propria with large numbers of CD3-positive lymphocytes. Eighty-five of the live WNPRC animals initially screened for SOBV in 2014 were euthanized for experimental end points or clinical illness between their screening and 3 May 2019. Sixty-nine (81.18%) of these animals were diagnosed by postmortem histological analysis with lymphocytic enteritis, colitis, or enterocolitis. The data from two animals were removed from this analysis due to confounding factors (one animal had severe tissue autolysis, and the other animal had B cell lymphoma of the small and large intestines).

Pegivirus infection was not found to be associated with an increased likelihood of developing lymphocytic enteritis in the small intestines (*p* = 0.779), colitis in the large intestine (*p* = 0.196), either a colitis or enteritis (*p* = 0.820), or an enterocolitis (*p* = 0.0798), or with lack of any lymphocytic disease (*p* = 0.904) (Figure 6). Sex was not associated with likelihood of the various disease states (*p* = 0.400, *p* = 0.912, *p* = 0.235, *p* = 0.812, and *p* = 0.235, respectively). SOBV status was likewise not associated with any other pathology (Appendix A).

## 4. Discussion

We describe the discovery of a novel simian pegivirus, the Southwest bike trail virus (SOBV), first identified in common marmosets diagnosed with lymphocytic enterocolitis. We show this pegivirus was prevalent in our colony during a period of increased incidence of lymphocytic enterocolitis and that it was less prevalent in a similar, clinically normal colony. The novel virus was not significantly associated with the likelihood of developing lymphocytic enterocolitis, though prevalence of the virus increased with increasing age in the common marmoset. With an average prevalence of 34%, SOBV was common throughout the WNPRC common marmoset colony.

Pegiviruses, the members of genus *Pegivirus* (*Amarillovirales*: *Flaviviridae*), have single-stranded, positive-sense RNA genomes and produce enveloped virions [84]. The first members of the genus were identified about 20 years ago [85,86], and since that time pegiviruses have been found in many animal populations [24,25,26,27,28,29,30,31,32,33,34,35,81]. Pegiviruses have never been shown to be causative agents of any disease or alteration in physiology [43,44,45,46,47,48,49,50,51,52,53,54,55,56,57,58,59,60]. Human pegivirus (HPgV) has been linked both to improved outcomes in HIV-1 infection [68,87,88,89,90,91,92,93,94,95,96,97,98,99,100,101,102,103,104] and to increased incidence of various types of lymphoma [105,106,107,108,109,110,111,112,113], though this remains controversial [114,115,116,117,118]. HPgV is considered likely lymphotropic [64,65,66,67], and evidence from in vivo and in vitro research suggests HPgV may affect T cell activation, signaling, proliferation and apoptosis, and CD4 or CD8 expression [68,69,70,71,72,73,74,75,76], and that it may be associated with a higher rate of host cell DNA damage [119] and genomic destabilization [110]. These effects on T cell functions may be a common pathway through which these viruses may cause lymphocytic diseases.

It is not known whether common marmosets are the natural host for SOBV or whether they acquired this virus from another species in captivity [120]. Other pegiviruses were discovered in wild common marmosets in the 1990s [121], but their prevalence has never been examined. The prevalence of SOBV in our captive common marmoset population was quite high compared to the prevalence of HPgV, which is found in about 1–4% of human populations [122,123,124,125,126,127,128,129], and compared to the prevalence of pegiviruses in captive chimpanzees (1–3%) [41]. SOBV is most similar to a pegivirus discovered in a three-striped night monkey (*Aotus trivirgatus*) [27], a species used in malaria research at other primate research facilities [130,131,132], indicating SOBV may have been introduced into common marmosets through contact in captivity. Interestingly, SOBV is highly similar to several variants of a bat pegivirus isolated from African straw-colored fruit bats (*Eidolon helvum*). Given that common marmosets and three-striped night monkeys are native to northern South America, this may indicate a South American bat species harbors a more closely related pegivirus and could have been the source of an interspecies spillover.

The routes of transmission of SOBV and of other simian pegiviruses have not been examined. HPgV transmission has been extensively studied and is known to occur efficiently through blood products or dialysis [36,45,133,134,135,136], intravenous drug use and needle sticks [133,137,138,139], sexual intercourse [133,137,140,141], and from mother to infant [133,142,143,144,145,146]. Captive common marmosets are typically housed in familial groups in shared cages and receive some vaccines and other medication by injection, and common marmosets frequently give birth to non-identical twins [3,4,5,22]. These animals thus have the potential to transmit SOBV through direct contact, sexual contact, birth, and medical injections or veterinarian manipulations. Defining mechanisms of transmission will be important in preventing infection and thereby allowing the study of this virus’ effects.

The high prevalence of this virus at the WNPRC raises important considerations about potential effects on common marmoset experiments. Facilities working with common marmosets should prescreen the animals to establish the pegivirus status of animals in research to account for potential confounding. Pegiviruses can replicate at high titers in a host for more than a decade [36,37,41,147]; thus, the length of time for which an animal has been continuously infected may also be relevant in potentially confounding study outcomes. Future investigations, perhaps involving the isolation of common marmosets for years at a time to follow the natural history of chronic pegivirus infection in these animals, could examine the long-term effects of infecting common marmosets with SOBV.

This study has several limitations. First, this study was observational in nature, as we did not want to risk infecting more marmosets in our research colony with an apparently transmissible and potentially harmful virus. This study design could not examine a causal link between viral positivity and the development of lymphocytic enterocolitis. Definitive establishment of causation would require demonstrating that animals infected experimentally develop the disease. Second, many animals in this study were concurrently enrolled in other WNPRC studies, and therefore some were euthanized earlier than would have occurred otherwise when those studies reached experimental endpoints. We chose to use this convenience sample as it allowed us to achieve a large study sample size in which to investigate a potential infectious contributor to an important and poorly understood cause of common marmoset mortality without disrupting other ongoing studies at the WNPRC. Third, not all of the animals initially screened were deceased at the time of this analysis, and future necropsies of these animals may contribute additional data concerning the likelihood of enterocolitis development. Finally, some animals in this study may have cleared the virus before the samples we tested were collected. Consequently, these animals could have been mistakenly classified as virus-naïve, and others may have acquired the virus after initial screening. Development of a SOBV-specific ELISA or other serodiagnosis tools would enable deeper appropriate analyses of SOBV infection rates both prospectively and retrospectively.

In summary, this work describes the discovery of a novel simian pegivirus and investigates its relationship with a widespread and devastating cause of common marmoset mortality. Our study lays the groundwork for the future development of a nonhuman primate model system using this natural infection as a potential model for studying the mechanisms of these enigmatic viruses and providing a greater understanding of their genus as a whole.

## 5. Conclusions

We discovered a novel pegivirus, SOBV, in common marmosets with lymphocytic enterocolitis. This novel virus had highly variable prevalence between two different colonies (34% prevalence in one and 6% prevalence in another). While SOBV was not found to be associated with pathology in this retrospective analysis, the difference in rates of infection among different colonies may affect the outcomes of studies done at different primate centers, and thus infection rates and prevalence should be monitored. Further investigations should probe the routes of transmission and the biological effects of experimental infection.

## Figures and Tables

**Figure 1 microorganisms-08-01509-f001:**
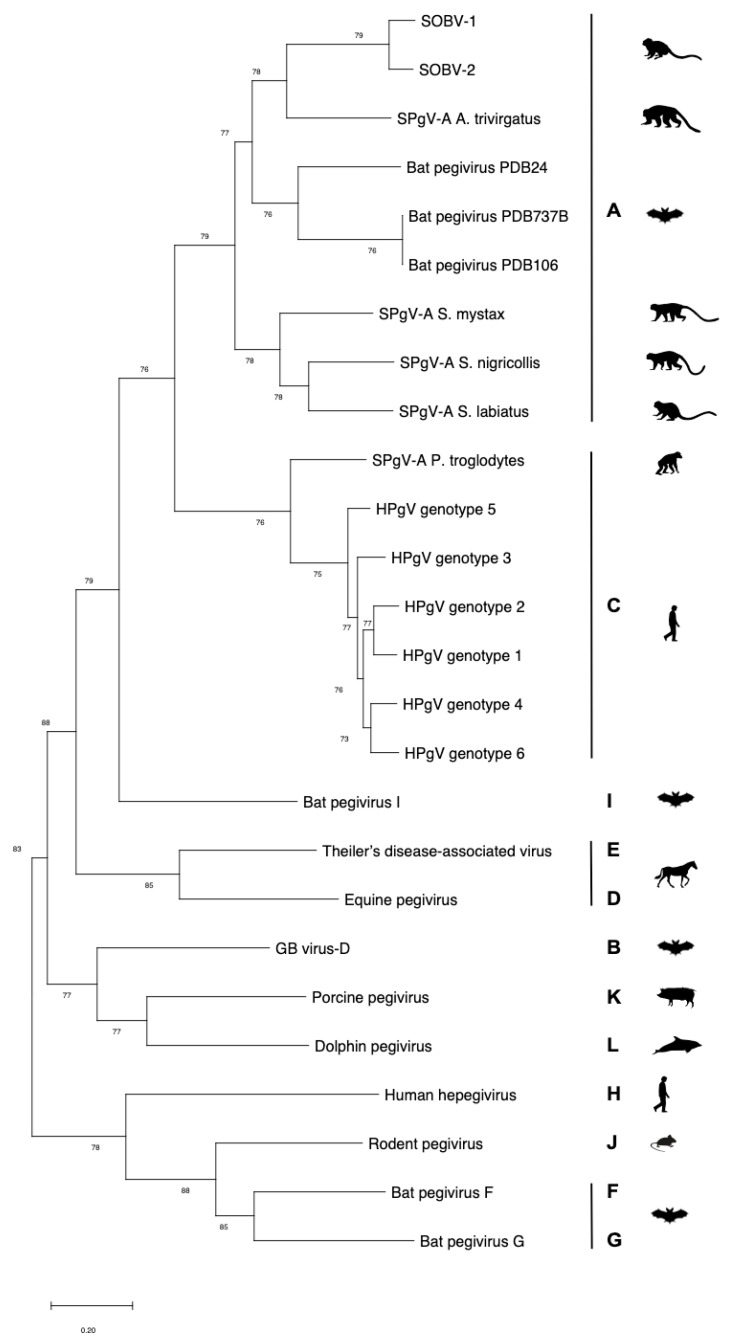
A phylogenetic tree of newly discovered pegivirus Southwest bike trail virus (SOBV) variants 1 and 2 shows it is most closely related to pegiviruses found in other New World monkeys and bats. We generated maximum likelihood trees using MEGA6.06 (1000 bootstrap replicates, GTR + I+γ model) from codon-based alignments (via MAFFT); bootstrap values of less than 70 were omitted. *Abbreviations*: HPgV = human pegivirus; SPgV = simian pegivirus.

**Figure 2 microorganisms-08-01509-f002:**
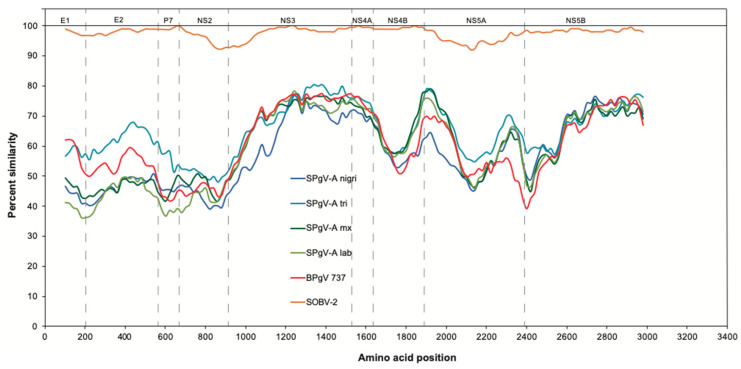
Sliding window similarity plots [79] show the relatedness of the amino acid sequences of SOBV-2 and other closely related pegiviruses to SOBV-1. Dashed vertical lines indicate the putative approximate start positions of inferred viral proteins, from left to right: E1, E2, P7, NS2, NS3, NS4A, NS4B, NS5A, and NS5B [80]. *Abbreviations:* SPgV-A nigri = GBV-A-like virus recovered from black-mantled tamarins (*Saguinus nigricollis*); SPgV-A tri = GBV-A-like virus recovered from three-striped night monkeys (*Aotus trivirgatus*); SPgV-A mx = GBV-A-like virus recovered from mustached tamarins (*S. mystax*); SPgV-A lab = GBV-A-like virus recovered from white-lipped tamarins (*S. labiatus*); BPgV 737 = bat pegivirus recovered from African straw-colored fruit bats (*Eidolon helvum*).

**Figure 3 microorganisms-08-01509-f003:**
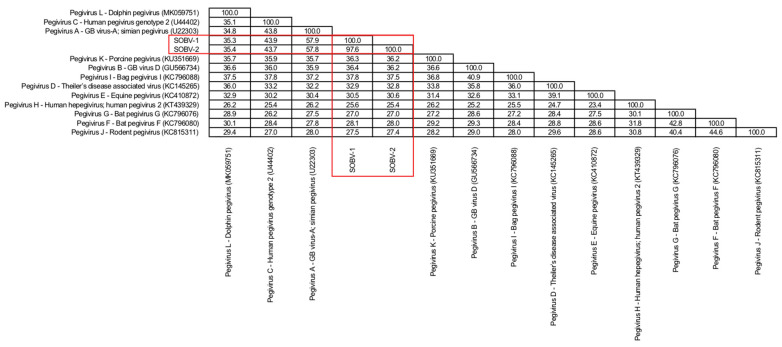
Sequence identity matrix based on amino acid alignment of the newly discovered SOBV-1 and SOBV-2 (red box) compared to members of the 11 recognized pegivirus species and of one proposed species [80]. (The classification of dolphin into species “Pegivirus L” has been suggested [81].)

**Figure 4 microorganisms-08-01509-f004:**
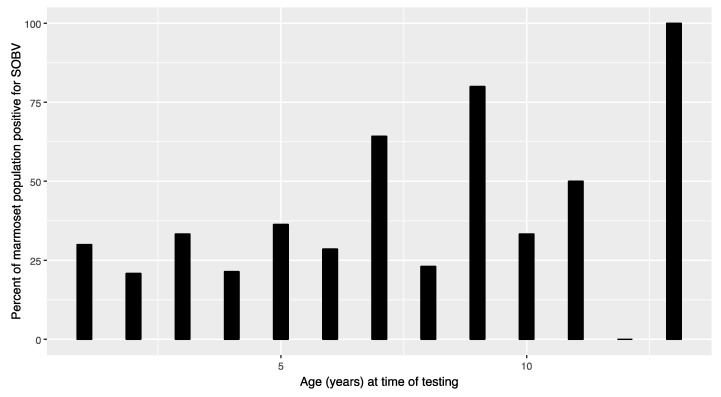
Prevalence of infection with SOBV in common marmosets at the WNPRC increases with age. One hundred forty-six live, clinically normal common marmosets in the WNPRC captive common marmoset colony were screened for SOBV using RT-PCR and deep-sequencing methods. The likelihood of infection with these viruses was significantly statistically associated with increasing age (*p* = 0.03237) using univariate logistic regression.

**Figure 5 microorganisms-08-01509-f005:**
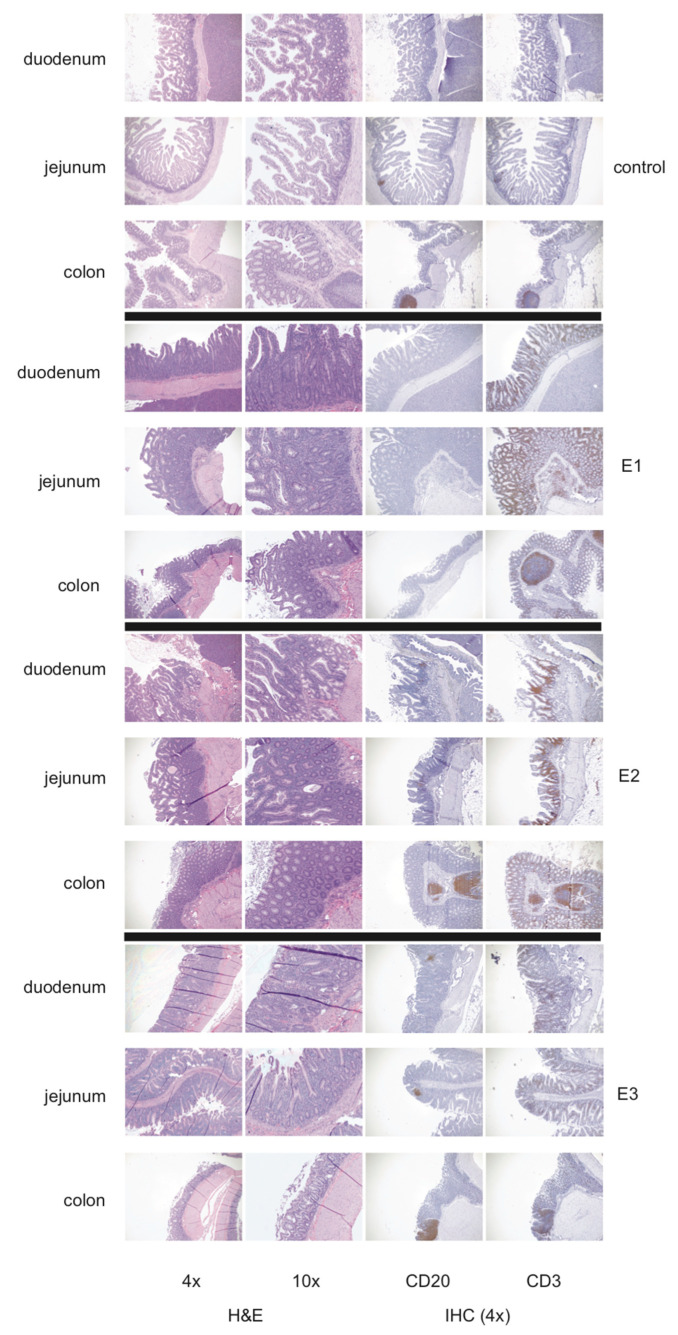
Representative photomicrographs show disruption of the normal architecture in the duodenum, jejunum, and colon by lymphocytic enterocolitis in common marmosets. Histology was performed upon intestinal samples from 85 common marmosets. Intestinal sections were stained with hematoxylin and eosin (H&E) and with B cell-specific and T cell-specific staining procedures (immunohistochemistry) with monoclonal antibodies to CD20 or CD79 (B cell markers) and CD3 (T cell marker), respectively. E1, E2, and E3 represent three different marmosets with lymphocytic enterocolitis. E1 has disease in the duodenum and jejunum that spares the colon, while E2 and E3 have disease in the duodenum, jejunum, and colon.

**Figure 6 microorganisms-08-01509-f006:**
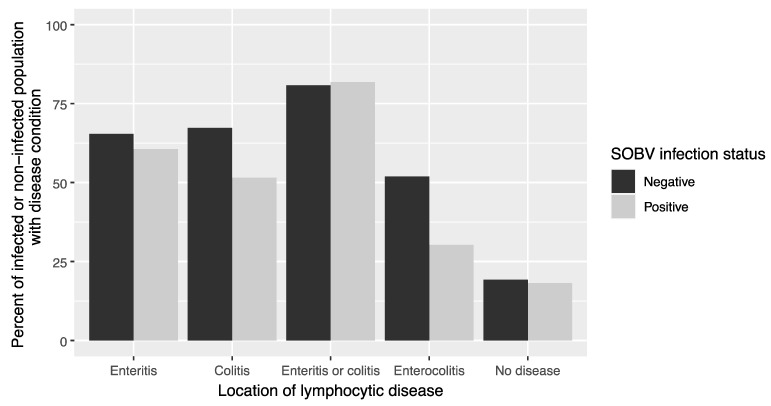
Infection with SOBV is not associated with the likelihood of developing lymphocytic enteritis, colitis, or enterocolitis. Eighty-five common marmosets at the WNPRC, which had been previously screened for SOBV by RT-PCR or deep-sequencing of plasma samples, were examined postmortem for histological evidence of lymphocytic enterocolitis. Pegivirus infection was not found to be associated with an increased likelihood of developing lymphocytic colitis (*p* = 0.196), enteritis (*p* = 0.779), either enteritis or colitis (*p* = 0.820), enterocolitis (*p* = 0.0798), or lack of lymphocytic disease (*p* = 0.904), using univariate logistic regression.

**Table 1 microorganisms-08-01509-t001:** SOBV status by age of 146 common marmosets at the WNPRC.

Age (Years) at Time of Screening	Total Number of Marmosets	Number of Marmosets Infected	Number of Marmosets Non-Infected
1	20	6	14
2	24	5	19
3	18	6	12
4	14	3	11
5	22	8	14
6	7	2	5
7	14	9	5
8	13	3	10
9	5	4	1
10	6	2	4
11	2	1	1
12	0	0	0
13	1	1	0

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
