# Peer review of "Discovery of a Novel Simian Pegivirus in Common Marmosets (Callithrix jacchus) with Lymphocytic Enterocolitis"

_microorganisms, 2020, doi:10.3390/microorganisms8101509_

Round 1
Reviewer 1 Report
In this manuscript, the authors described newly discovered pegivirus identified form common marmosets diagnosed with lymphocytic enterocolitis and revealed that this novel pegivirus is not associated with lymphocytic enterocolitis. The manuscript is well-written and the presented results are well-organized with limited samples and research conditions. Though further characterizations of the novel pegivirus are needed, data presented it this manuscript is worth to publish for not only virological insight but also to enlighten the persistent infection of the pegivirus for the researcher using common marmosets.
Comments:
- The Gene accession numbers of each viral sequence used in Fig 1 and 2 need to be listed in this manuscript.
- Did the authors used amino acid sequences from all open reading flame of pegiviruses or specific amino acid sequences of t viral protein?
Author Response
Point 1: The Gene accession numbers of each viral sequence used in Fig 1 and 2 need to be listed in this manuscript.
Response 1: Reviewer 1 noted we had not included the GenBank accession numbers used in Figures 1 and 2. We are grateful to Reviewer 1 for noticing this; we have included these accession numbers in the methods (lines 169-181 and 185-189).
Point 2: Did the authors used amino acid sequences from all open reading flame of pegiviruses or specific amino acid sequences of t viral protein?
Response 2: Reviewer 1 asked whether we used the amino acid sequences from all open reading frames in the pegiviruses analyzed. Pegiviruses have only one open reading frame, so we did use all open reading frames for each pegivirus.

Reviewer 2 Report
The manuscript reports the identification of a previously unknown pegivirus from the blood of common marmosets with lymphocytic enterocolitis by using NGS approach. Then authors screened their colony to detect the diffusion of the virus and to evaluate possible linkage between the infection and the presence of pathologies, notably lymphocytic enterocolitis.
Manuscript is clear and well written.
Materials and Methods section should be improved. In particular:
- Lines 87 to 96, the text should be removed;
- line 187, replace “Postmortem” with Post mortem
- lines188 -196, more details should be provided about the criteria adopted to define the cases of lymphocytic enterocolitis, was standardised protocol developed and applied on the animal during necropsy?
Author Response
Point 1: Lines 87 to 96, the text should be removed; line 187, replace “Postmortem” with Post mortem.
Response 1: Reviewer 2 noted some editing needed in the methods. We are grateful to Reviewer 2 for noticing these areas, and we have made the changes suggested (lines 87 and 212).
Point 2: lines188 -196, more details should be provided about the criteria adopted to define the cases of lymphocytic enterocolitis, was standardised protocol developed and applied on the animal during necropsy?
Response 2: Reviewer 2 requested more details be added to the methods concerning how lymphocytic enterocolitis is defined. We have expanded this part of the methods with additional details (lines 213-218). We are grateful to Reviewer 2 for asking for clarifications that will be useful to other readers of this work.
